# The Pattern of Substance Use among People Who Use Drugs (PWUD) Receiving Treatment at University Malaya Medical Centre (UMMC) during the COVID-19 Pandemic and the Associated Factors

**DOI:** 10.3390/healthcare10081366

**Published:** 2022-07-23

**Authors:** Amir Zulhilmi bin Yahaya, Anne Yee, Ahmad Hatim Sulaiman

**Affiliations:** 1Department of Psychological Medicine, University of Malaya, Kuala Lumpur 59100, Malaysia; annyee17@um.edu.my (A.Y.); hatim@um.edu.my (A.H.S.); 2University Malaya Centre of Addiction Sciences (UMCAS), Faculty of Medicine, University of Malaya, Kuala Lumpur 50603, Malaysia

**Keywords:** COVID-19, anxiety, depression, pandemic, coping mechanism, isolation, PWUD (people who used drugs)

## Abstract

There was a change in the pattern of substance usage among people who use substances during the COVID-19 pandemic period. This study aims to determine the effects of the COVID-19 pandemic on the pattern of substance usage among people who use drugs (PWUD) receiving treatment at the University Malaya Medical Centre (UMMC) as well as levels of anxiety and depression together with coping mechanisms and the factors affecting the pattern of substance use during COVID-19 pandemic period. A cross-sectional study was applied. The questionnaire used was the Mini-European Web Survey on Drugs (EWSD): COVID-19, Hospital Anxiety and Depression Scale (HADS), and Brief COPE Scale. In total, 130 PWUD were recruited. Of the participants, 36.2% of PWUD had not used/stopped the usage of illicit drugs/alcohol, 26.2% increased their usage, 20% decreased, and 14.6% used the same amount of illicit substances/alcohol during the COVID-19 pandemic period/restrictions. In addition, 28.5% of PWUD had an increased intention to seek professional support for drug counseling/treatment during the COVID-19 pandemic period. The prevalence anxiety and depression symptoms in PWUD according to HADS was 33% and 41.5%, respectively, with depression (*p* = 0.05) and isolation status (adjusted OR = 2.63, *p* < 0.05) being associated with an increase in alcohol/illicit substance use during the COVID-19 pandemic. PWUD who had increased their intention to seek professional support had significantly higher odds (adjusted OR = 4.42, *p* < 0.01) of reducing their alcohol/illicit substance use during the COVID-19 pandemic period. There were increased odds of maintaining alcohol/illicit substance usage among PWUD who practiced dysfunctional coping (adjusted OR = 3.87, *p* < 0.025) during the COVID-19 pandemic period. In conclusion, depression, isolation status, dysfunctional coping, and intention to seek professional support affected the pattern of alcohol/illicit substance use during the COVID-19 pandemic period. Strategies, substance rehabilitation/counseling, and proper mental health screening and the associated risk factors must be emphasized to prevent a further epidemic of substance use during the pandemic.

## 1. Introduction

COVID-19 (Coronavirus Disease 2019) was announced by World Health Organization (WHO) as a global pandemic, and infections continue to increase around the globe. It first originated in Wuhan, China, in November 2019 and spread across the world at a rapid rate. Some of the common strategies as advised by WHO in slowing the rate of the infection included preventive measures such as the limitation of movement in the infected area, interruption of human–human transmission using social distancing, early identification and isolation, and providing appropriate care for patients. During this pandemic period, information regarding the virus spread faster and more extensively compared to the SARS outbreak in 2003 due to modernization and globalization. This may have resulted in increased public fear, panic, anxiety, and stress, especially in people who are more susceptible to psychological distress [1].

The virus has caused a certain vulnerable population to have higher rates of morbidity and mortality; for example, substance use disorder patients and people who use drugs [1]. It also has made proper rehabilitation, diagnosis, and treatment of patients with substance use disorder difficult [2]. This may be due to the responses towards the pandemic, including movement restriction orders, total lockdowns, and quarantine [3]. In addition to these responses, the COVID-19 period has negatively affected people, especially those with prior psychological problems, and has resulted in maladaptive behavior, psychological distress, and poor coping mechanisms. Looking back at history, during the time of SARS (Severe Acute Respiratory Syndrome), a study conducted in Beijing, China on the psychological impacts of the outbreak/epidemic/pandemic period shows that around 40% of those studied suffered from post-traumatic stress disorder, even 3 years after the outbreak [4]. The association between adverse life events such as an outbreak and the brain stress system plays a leading role in addiction. People who use drugs (PWUD) are much more vulnerable to stress, which may cause relapse among existing or previous drug users [1].

A study in China from January to February 2020 indicates that about 54% of respondents rated the psychological impact of COVID-19 as moderate to severe, which may impact the pattern of substance usage of people who already used substances before the pandemic period [5]. Based on public and health provider observation, there is informal evidence that there were changes in patterns and behavior of people who use drugs during this pandemic period as a result of the government responses towards the escalation of COVID-19 [3].

The lockdowns, isolation, and restriction of movement implemented by governments around the globe may cause an increase in substance use due to feeling distressed and anxious, as reported by a previous study during the SARS outbreak [6], or drug users may use this time to free themselves from the substance as it may be difficult for them to access their respective substances.

From multiple studies’ perspectives, there are many psychosocial factors contributing to substance and alcohol use relapses among users, including low self-efficacy, attitude, poor knowledge of drug abuse, poor family and social support, peer pressure, unemployment, as well as lack of well-established drug rehabilitation centers [7]. These could be some of the confounding factors that are associated with a change in the patterns of substance abuse or relapse. However, the limitation of these studies is the lack of attention to the individual characteristics or mental well-being that may exacerbate the possibility of relapse in individuals who use substances.

Other consequences of COVID-19 were the devastating economic effects, which resulted in income and employment losses for millions of people around the globe. The direct and indirect outcome of this was the emergence and increase of both psychological symptoms and disorders, including stress, anxiety, depression, substance use/abuse, and many others. Moreover, COVID-19 as well as other pandemics are postulated to hinder substance use disorder treatment, causing probable growth in withdrawal symptoms and relapses among substance users worldwide [8]. According to the Anxiety and Depression Association of America, 20% of Americans who have an anxiety or depressive disorder also have a substance use disorder, and 20% of those who have a substance use disorder also have an anxiety or depressive disorder. Substance users may also increase or change their patterns of substance use as a coping mechanism due to COVID-related worry or anxiety [8].

Furthermore, the increased overdose mortality and morbidity due to substance use resulting from possible psychological implications of COVID-19 itself e.g. social distancing, isolation or quarantine would cause a further increase in the risk of respiratory distress and lung disease which complicates the COVID-19 cases among substance users [9]. Some of the COVID-19 complications such as acute respiratory distress syndrome (ARDS), renal failure, and death were increased in people who uses substances, particularly those using alcohol and drugs such as opioids, benzodiazepines, and methamphetamines which damage the lungs and impaired the respiratory system [10]. 

As for the Malaysian setting, to date (13 April 2022), there have been (4,342,559) positive COVID-19 cases reported nationwide, with (35,341) people dying due to the virus. It has been observed that the pandemic period has affected people psychologically, especially those in contact with or contracting the virus. There are ongoing studies regarding the mental health effects of the COVID-19 pandemic in Malaysia as well as substance use patterns among Malaysians who uses substances. Currently, there is limited data regarding the number of substance user relapses or changes of patterns during the pandemic period. Hence, it is important to identify these changes among substance/drug users to initiate proper management plans and rehabilitation at the University Malaya Medical Centre (UMMC) setting during this global pandemic or any prolonged containment in the future.

In this study, we aim to study the effects of the COVID-19 pandemic on patterns of substance usage among people who use drugs (PWUD) at the University Malaya Medical Centre (UMMC). Specifically, we looked at the association of isolation status, anxiety, and depression on the pattern of substance use during the pandemic as well as the different coping mechanisms used.

## 2. Materials and Methods

A cross-sectional study was commenced amongst PWUD at the University Malaya Medical Centre (UMMC). The participants were recruited via convenience sampling and screened by trained doctors in the clinic for inclusion and exclusion criteria before inclusion in this study. Subjects were patients under psychiatric follow-up in UMMC or were referred to the psychiatry department at UMMC. However, given there were restrictions on face-to-face interactions and physical distancing during the outbreak of COVID-19, subjects were recruited via an online platform using Google Forms. The online questionnaire used was the Mini-European Web Survey on Drugs (EWSD): COVID-19 (Both English and Malay Translation versions). In this study, isolation status was grouped into the following three categories:Physical isolation = (e.g., avoiding public transport and social gatherings, working/studying from home);Home isolation = (i.e., government asked people to stay in isolation at home);Home quarantine = (tested positive for/close contact with COVID-19 and stayed at home).

Potential subjects also were assessed regarding their depression and anxiety levels during COVID-19 using the Hospital Anxiety and Depression Scale (HADS) (Both English and Malay Translation version) and coping mechanisms using the Brief COPE Scale via an online platform using Google Forms. The information given was kept confidential.

### 2.1. Inclusion Criteria

Subjects who are capable of understanding and reading Malay or English;Subjects who are above the age of 18;Subjects who gave consent regarding the participation in this study;Subjects who use substances/alcohol.

### 2.2. Exclusion Criteria

Subjects who did not give consent for the study;Subjects who are not capable of reading and understanding Malay or English;Subjects who are physical and mentally unstable (not in a severe acute episode of psychiatric illnesses).

Socio-demographic data and information regarding the patterns of substance usage were gathered using the preconstructed questionnaire Mini-European Web Survey on Drugs (EWSD), designated by the European Monitoring Centre For Drugs and Drug Addiction (EMDDCA), used with permission and translated and validated using cognitive debriefing and pilot testing among more than 10 bilingual trainees in the Master’s of Psychiatry program. Depression and anxiety were evaluated via English and Malay translation of the Hospital Anxiety and Depressive Scale (HADS). Participants’ coping mechanisms were measured using English and Malay translations of the Brief COPE Scale. All tools were validated to be used in the Malaysian setting, except for EWSD. This study was approved by the Medical Ethical Committee of UMMC (MREC: 2020420-8537).

### 2.3. Measurement Tools

#### 2.3.1. Mini-European Web Survey on Drugs (EWSD): COVID-19

The Mini-European Web Survey On Drugs (EWSD): COVID 19 aims to gather information about how patterns of drug use may have changed in Malaysia due to COVID-19. It is a series of questions about substance usage, especially during the COVID-19/Movement Restriction Order period. Permission from EMDDCA to use the EWSD for this study was obtained before the study.

There were no specific validity studies done for the Mini-European Web Survey on Drugs (EWSD): COVID-19; however, there were reliability and validity studies conducted for the previous EWSD Questionnaire. For most prevalence items with acceptable sample sizes, test–retest reliability was judged moderate to good (reliability coefficients between 0.55–0.87). The forward translation from the original language to another language (Malay language) was done by two independent translators, and the finalized version of EWSD: COVID-19 (Malay language version) was produced after pilot testing to determine cognitive debriefing.

#### 2.3.2. Hospital Anxiety and Depression Scale (HADS)

The HADS is a 14-item scale that requires respondents to give a verbal response, which is scored as an index of the severity of anxiety or depression. The scores are then summed to produce two subscales corresponding to Anxiety (HADS-A) and Depression (HADS-D). As well as the subscale totals, an overall total can be derived to indicate the level of psychological distress. Each item is scored 0–3, with the maximum score of 21 for each subscale [11]. A lower cut-off score of 8 was applied in this study to include a significant fraction of the Malaysian population with anxiety and depression [12], with a sensitivity and specificity of 93.2% and 90.8%, respectively. The Malay version of this scale had a Cronbach’s alpha of 0.87 [13].

#### 2.3.3. Brief COPE Scale

The Brief COPE is a self-reporting questionnaire; it consists of 28 statements that assess a particular way of coping. The Brief COPE measures how frequently a person has been doing certain activities to cope with stressful situations in daily life. The items of the questionnaire were rated under 4 categories of responses. The recommended scoring method was 1 for the lowest frequency and 4 for the highest frequency—i.e., 1–2–3–4. The minimum and maximum Brief COPE total scores were 2 and 8, respectively [14]. The recommended scoring of this scale involves dividing it into 14 subcategories or styles of coping mechanisms, with three main categories: problem-focused, emotion-focused, and dysfunctional coping [15].

### 2.4. Statistical Analyses

The Statistical Package for Social Science (SPSS) version 25.0 (IBM Corporation, Armonk, NY, USA) was used to analyze the data in this study. Descriptive statistics were used to demonstrate the properties of the study population. Categorical variables were presented in frequency and percentage. Means and standard deviations were used to present the continuous variables. Chi-square and Fisher’s Exact Test were used to determine associations between sociodemographic elements, isolation status, depression, and anxiety status, as well as coping strategies associated with the increased, reduced and unchanged usage pattern of substances, used for the bivariate analysis. Covariates with *p* values of less than 0.25 from the bivariate analysis were tested further via multivariate analysis [16].

## 3. Results

### 3.1. Socio-Demographic Data

Table 1 demonstrates the socio-demographic information of the study subjects. In total, 130 PWUDs participated in the study. The mean age of the 130 subjects enrolled in this study was 45.98 (SD ± 9.78). The majority of the participants were male (83.8%, *n* = 109), while 16.1% (*n* = 21) were females. The majority of the participants resided in city areas (96.2%, *n* = 125), while only (2.3%, *n* = 3) lived in town areas and (1.5%, *n* = 2) lived in village areas. Regarding the different COVID-19 isolation statuses amongst the study population, the majority had both physical isolation and home isolation (48.5%), followed by only physical isolation (26.6%) and physical isolation, home isolation, and home quarantine (6.2%). Participants who had physical isolation, home isolation, and home quarantine tested positive for COVID-19 (6.2%). Most of the participants used multiple substances (42.3%, *n* = 55), followed by heroin (19.2%, *n* = 25), alcohol (10%, *n* = 13) and amphetamine-type stimulants (8.5%, *n* = 11) prior to the COVID-19 pandemic period.

### 3.2. General Use of Illicit Substances/Alcohol Pattern during COVID-19 Restrictions

Table 2 demonstrates the general use of illicit substance/alcohol patterns during COVID-19 restrictions: (36.2%, *n* = 47) of people did not use/stopped taking the illicit substances/alcohol during COVID-19 restrictions, whereas (26.2%, *n* = 34) increased their usage, and (20%, *n* = 26) used lower amounts of illicit substances/alcohol. The percentage of people who used the same amount of illicit substances/alcohol during COVID-19 restrictions was (14.6%, *n* = 19). As a whole, (82.3%, *n* = 107) of the participants changed their illicit substance/alcohol usage pattern during the COVID-19 restrictions, while (17.7%, *n* = 23) of them did not.

### 3.3. Substances Used during COVID-19 Pandemic Period and the Pattern of Usage for Each Substance

During the COVID-19 pandemic period and restrictions, there was a change of pattern in term of types of substances used as well as their frequencies of usage. In Table 3, for multiple substance use, (14.6%, *n* = 19); for alcohol, heroin, and amphetamine-type stimulants, (12.3%, *n* = 16) of participants used each of the substances; and for cannabis (6.9%, *n* = 9). Below is the comparison of substance use prior to and during the COVID-19 pandemic period and in Table 4, the pattern of usage for each of the substances. 

### 3.4. Changes in Purity, Price, and Amount of Illicit Substances/Alcohol Obtained since the Outbreak of COVID-19

Since the outbreak of COVID-19 worldwide, particularly in Malaysia, the study population noticed changes in term of purity, price, and amount of illicit substances/alcohol available on the market, which shown in the Table 5 below. For the purity of the substances, generally, (36.2%, *n* = 47) of the study population found that the purity had decreased since the COVID-19 outbreak, (18.5%, *n* = 24) found it to be the same, and (0.8%, *n* = 1) found it to be increased, while the rest of the respondent answered, ‘others’ (44.6%, *n* = 58).

In terms of the general price of the substances, (40%, *n* = 52) of the population found it to be the same, (10%, *n* = 13) found it to be higher, and (5.4%, *n* = 7) found it to be lower since the outbreak of COVID-19, while the rest of the respondent answered, ‘others’ (44.6%, *n* = 58). For the general amount of substances available, (44.6%, *n* = 58) found it to be the same, (10.8%, *n* = 14) found it to be decreased, (0.8%, *n* = 1) found it to be increased, while the rest of the respondent answered, ‘others’ (43.8%, *n* = 57).

### 3.5. Intention to Seek Professional Support (Counselling or Drug Treatment) to Reduce or Abstain from Use of Illicit Drugs since COVID-19 Restrictions

Figure 1 demonstrates the intention to seek professional support (counselling or drug treatment) to reduce or abstain from the use of illicit drugs since COVID-19 restrictions, most PWUD had no change in their intention (63.8%, *n* = 83), followed by a slight increase (23.1%, *n* = 30) and a strong increase (5.4%, *n* = 7).

### 3.6. Anxiety, Depressive Symptoms, and Coping Strategies amongst PWUD

From our study, we found that the prevalence of depressive symptoms was 41.5%, (*n* = 54/130), and the population who reported anxiety symptoms was 33% (*n* = 43), while 25.5%, (*n* = 33) did not experience anxiety and depressive symptoms.

#### 3.6.1. Anxiety and Depression Score

The mean score for anxiety was 5.03 (SD = 4.00). From 130 people who used drugs/alcohol, 66.9% (*n* = 87) reported a normal score, 21.5% (*n* = 28) reported borderline abnormal anxiety, and 11.5% (*n* = 15) reported an abnormal anxiety score.

From this study, the mean depression score was 6.44 (SD = 3.24). From 130 people who used drugs/alcohol, 58.5% (*n* = 76) reported a normal score, 33.8% (*n* = 44) reported borderline abnormal depression, and 7.7% (*n* = 10) reported an abnormal depression score.

#### 3.6.2. Descriptive Analysis of Brief-COPE

The mean score for each domain of coping strategies according to Brief-COPE employed by the people who used drugs/alcohol is given in the table below. The coping strategies are further subdivided into problem-focused coping (active coping, planning, instrumental support) (mean = 14.97 + 3.62), emotion-focused coping (acceptance, positive reframing, religion, emotional support, humor) (mean = 24.83 + 6.00), and dysfunctional coping (self-distraction, venting, self-blame, behavioral disengagement, denial, and substance use) (mean = 25.17 ± 4.19). From these results, most of the PWUD used dysfunctional coping strategies, as the mean was higher in the dysfunctional coping group (25.17 + 4.19).

#### 3.6.3. Correlation between Anxiety and Depression Scores with Different Coping Mechanisms among People Who Used Drugs/Alcohol (PWUD)

For the correlation between anxiety and depression scores with different coping mechanisms among people who used drugs/alcohol (PWUD), bivariate Pearson correlation tests were conducted. Each of the different coping mechanisms/strategies were tested for their correlation with both anxiety and depression scores. In Table 6 below, there was a statistically significant inverse relationship between problem-focused coping strategies and anxiety scores (R = 0.18, *p* value < 0.05).

The factors that are associated with increased alcohol/illicit substance usage were analyzed using simple and multivariable logistic regression. The variables/factors with a *p* value of <0.25 in the bivariate analysis were used for the analysis (Hosmer Jr, Lemeshow, & Sturdivant, 2013). For the sociodemographic variables, specifically looking at the gender and age, although not statistically significant, this study showed that male gender increases the odds of alcohol/illicit substance use compared to female gender (Crude OR = 1.96, *p* = 0.18). As for the age factor, increasing age showed as a protective factor against increasing alcohol/illicit substance use during the COVID-19 pandemic period.

This simple logistic regression model in Table 7 also showed that being in isolation increases the odds of alcohol/illicit substance use during the COVID-19 pandemic period at least two-fold, nearing the statistical significance level (Crude OR = 2.11, *p* = 0.08).

A multivariable logistic regression was performed to study the association between the variables in the simple logistic regression on increased alcohol/illicit substance usage during the COVID-19 pandemic period. From Table 8, the results show that those who were in isolation during the COVID-19 pandemic period increased their substance usage 2.63 times more than those who were not in isolation (*p* < 0.05), after the model was adjusted for age, gender, and place of living.

A multivariable logistic regression was performed to study the association between the variables in the simple logistic regression of reduction of usage of alcohol/illicit substances usage during the COVID-19 pandemic period. From Table 9, the results show that those who had an increase in intention to seek professional support (drug counselling/treatment) would reduce their substance usage 4.42 times more than those who had no increased intention to seek professional support (*p* < 0.01), after the model was adjusted for age, gender, and depression status.

A multivariable logistic regression was performed to study the association between the variables in the simple logistic regression on no change of pattern of alcohol/illicit substance usage during the COVID-19 pandemic period. From Table 10, the results show that those who were using dysfunctional coping strategies would be 3.87 times more likely to not change their substance usage than those who were not using dysfunctional coping strategies (*p* value = 0.025), after the model was adjusted for age, gender, and depression status.

## 4. Discussion

The COVID-19 pandemic has had devastating effects on the world in terms of physical and mental health, the economy, and many other elements. The world was basically ‘halted’ during the pandemic era, which affected different populations in many ways. [17]. Some of the populations, such as people who use substances, are much more vulnerable and at risk of contracting the disease compared to other populations. The negative impacts of the restrictions and pandemic response measures toward people who use drugs (PWUD) were devastating. Due to the reduction of drug supply and production worldwide, many dealers reduced the purity of the substances and increased the cost, which could result in poisoning, drug withdrawal, serious health problems, and increased risk of overdose [18]. Based on a recent study by the Centers for Disease Control and Prevention, there has been a 13% increase in substance use among the American population in recent years [19]. In addition, about 31% of adults in the United States suffer from anxiety or depression; at least 26% suffer from stress-related disorder symptoms and 11% have strong suicidal ideation [20], which also notable from the study done by SAMHSA [21]. 

Looking at the types of substances used before COVID-19, most of the respondents of this study used multiple substances (42.3%), followed by heroin and other types of substances. This prevalence was comparable to other studies, where multiple substance use was reported in 45.8% of participants who meet the criteria for substance use disorder [22].

For the general pattern of illicit substance/alcohol usage during COVID-19 restrictions, most of the respondents did not use or stopped taking the illicit substance/alcohol during the COVID-19 restriction period, followed by increased and decreased usage. These were similar results to other similar studies, which reported most of their web survey respondents declared using the same amount (21.3%) and increased usage of substances (21.2%). However, the same study also reported different percentages of people who decreased or stopped their usage. These findings might suggest some of the users were adapting their usage to a new restrictive environment due to the pandemic itself, although this is not applicable to the whole population [23].

In this study, we also looked at the pattern of use for the specific substances most commonly used by our respondents. The four most common substances were heroin, alcohol, amphetamine-type stimulants, and cannabis. For heroin use, we can see most of the users stopped their usage during the COVID-19 pandemic period (20.8%). Although these numbers and percentages were not comparable to other studies, the ratio and pattern for both categories are noted in a study from Luxemburg as well [23]. The reason behind this is most likely the scarcity of heroin during the pandemic, due to border restrictions as well as confinement policies [24].

For the pattern of alcohol use, most did not change their pattern of use (10%), followed closely by an increased intake of the same substance. This was in line with findings by various studies. For example, a study in Italy within a vulnerable group of substance users suggested that with COVID-19 and its restrictions, the shifting towards alcohol use and maintaining its use were inevitable during this pandemic times, as alcohol was one of the most readily available substances [25]. Similarly, a study in Amsterdam of changing patterns of substance use during COVID-19 reported similar trends [26]. 

The amphetamine-type stimulant (ATS) use pattern was also affected in some ways due to the COVID-19 pandemic. Most maintained their usage (13.8%) during this time, and 7.7% increased their usage. However, interestingly, among all other substances, 1.5% of respondents started using ATS during the COVID-19 pandemic. This was in line with other similar surveys done by the United Nations, which found that ATS and other synthetic drug users maintained usage; however, shifts occurred from traditional illicit drugs (that required illegal import) towards synthetic drugs, especially ATS, due to the availability of this kind of drug not being affected by the COVID-19 pandemic [27].

Some changes were also noted in the pattern of cannabis use during this pandemic era. In this study, 13.8% of respondents reported they had stopped their usage during the COVID-19 pandemic, which is most of the users. This result was comparable to another study in Amsterdam, where around 16.3% of users stopped using cannabis during the COVID-19 period. However, the discontinuation of cannabis was less common compared to other drugs [28]. A study done in Canada among cannabis users found that around half of the users increased their usage, which contradicts our findings. This might be a result of the availability of cannabis in Canada, where the legality of cannabis is much more accepted as compared to Malaysia [29].

As the COVID-19 pandemic hit the world, it caused a major stir in the drug market around the globe, especially in the countries that were most affected by the outbreak, including Malaysia. The stirs in the drug market due to restrictions and COVID-19 itself caused changes in purity, price, as well as the amounts of illicit substances/alcohol obtained since the outbreak of COVID-19. Most of the respondents found that the purity of substances generally decreased since the COVID-19 outbreak (36.2%). In terms of the price of the substances obtained, most of the users found it to be the same (40%), although the purity was lower. As for the general amount of substances obtained at the same price, the majority (44.6%) reported it to be the same amount as the pre-pandemic period. These results were noted to be different from other parts of the world, for example, Luxemburg, where most of the respondents to a web survey reported that the purity of the substances was the same (32.4%) but the price was higher (26.2%) and the quantity was the same (30.5%). This shows the drug market in this country adapted quickly during the COVID-19 pandemic period; the same pattern can be observed in other parts of Europe, for example, Switzerland [22]. This also was noted in Southeast Asia, where the drug market, especially methamphetamines, was not greatly affected, resulting in little change in price; however, the purity of the substances was affected, especially in the initial part of the pandemic, due to availability and mobility restrictions [27].

The depression and anxiety symptoms among people who use drugs/alcohol were also noted to be prevalent in this study. These were comparable to other studies, which also indicate a prevalence of anxiety and depression of around 41.1% within the community [30]. A study published by Czeisler et al. also suggested that anxiety and depression are prevalent among substance users, reported at around 42% [19].

As for the coping skills and strategies, most of the respondents adapted dysfunctional coping strategies, followed closely by emotion-focused and problem-focused coping strategies. As in other studies, most substance users used some sort of pathological way of coping, specifically dysfunctional coping [31]. Poor emotional regulation is one of the main causes of dysfunctional coping among substance users, and this association works bidirectionally [32].

Based on this study, there was no significant association between sociodemographic status and change in pattern of substance use. However, the association was prominent and even significant between sociodemographic data and increase in substance use during the COVID-19 pandemic period. There was a significant association between the age of the respondents and an increase in substance use, with a *p*-value of 0.04 (<0.05) in the bivariate analysis. According to a study by the National Institute of Drug Abuse in 2020, there is a significant association between age and increased substance use among substance users [33]. Substance use is also prevalent for users aged around 35 years and above, partly due to adulthood roles, previous use, and experiences with advancing age [34].

However, in our multivariate analysis, advancing age was protective against the increase in alcohol/illicit substance usage among PWUD in the UMMC. A postulated hypothesis for this is, as the substance users grow older, they may develop a sense of guilt as well as multiple medical issues that may prohibit and prevent them from increasing their substance usage. The prevalence of substance users among older adults is also quite low compared to the previous generation, according to several studies [35]. However, several risk factors may increase the likelihood of them increasing or maintaining use in older age, such as comorbid mental health disorders, coping styles, and social factors [35].

Another notable sociodemographic factor associated with the increased usage of substances is isolation status, which was associated with a three-fold increased risk of increasing substance use during this COVID-19 pandemic period. There are significant data to support this association; a study involving US national data suggests a strong association between isolation during COVID-19 and prevalence or increase in substance use [36]. Isolation and solitary confinement during the pandemic also affected people who use drugs/alcohol, which then further changes their substance use pattern, specifically, increasing their usage [37].

This study found that there is a significant association between depression status and the increase in illicit substance/alcohol use during the COVID-19 pandemic period/restrictions but not with anxiety. Around 25% of people with alcohol addiction and 50% of people with substance addiction in the United States have co-morbid depression. Anxiety disorders can be identified in around 25% of cases of alcohol dependence and 43% of cases of drug dependence. Alcohol dependency may be accompanied by co-morbid mental illnesses, such as severe anxiety or depression, which can have a poor influence on the quality of life, functioning, and ability to react to treatment [38].

For the association between coping strategies and the change of pattern and increase substance usage, interestingly, our study found there was no significant associations. Although there are many studies suggesting that there is an association between dysfunctional coping mechanisms and an increase in substance/alcohol use, other studies found one-sixth of individuals (17.4%) used at least one substance to cope with stress regularly, showing the prevalence of substance-use coping in middle-aged and older persons. However, there was no link between frequent alcohol-use coping and self-rated health, which was in line with earlier research by Lin et al. [39].

In this study, we found there was a significant correlation between coping skills, particularly problem-focused coping, with anxiety and depression status among people who used drugs/alcohol (PWUD) in the UMMC. Other studies have shown that denial, behavioral disengagement, self-blame, self-distraction, drug use, acceptance, positive reframing, active coping, and seeking emotional support were all found to be strongly linked to anxiety, concerns, and sadness. The study showed regulating stressful emotions (emotion-focused coping strategies such as passive and active avoidance, escaping, seeking social support, and positively reappraising the stressor) and managing the problem that is causing the distress are two widely recognized major functions of coping (problem-focused coping strategies such as planning how to change the stressor, seeking practical or informational support, and confronting the stressful situation). Problem-focused coping strategies have been linked to improved adjustment, while emotion-focused strategies have been linked to poor adjustment [40]. This is similar to the findings in our study, which showed emotion-focused strategies linked with more depressive states and vice versa.

Specifically, the association between the dysfunctional coping strategies and no change in illicit substance/alcohol usage was noted to be significant in our findings. This shows that people who used substances with dysfunctional coping mechanisms tended to continue their usage patterns, especially during the COVID-19 pandemic period. The people who use dysfunctional coping mechanisms tend to not change their usage patterns. As mentioned earlier, some PWUD reduced or even stopped their usage due to multiple reasons, including coping with the pandemic situation and difficulties in obtaining certain substances. However, people that exhibited dysfunctional coping continued their usage despite many restrictions imposed by government policies. The severity of drug abuse was not found to be related to coping style. Furthermore, based on a study by Ingmar et al., after three months of inpatient substance misuse treatment, maladaptive coping methods declined, while more adaptive coping styles remained steady for another three months [41].

It was also noted that the intention to seek professional help and support for drug treatment/counselling was associated with a reduction in substance usage itself during the COVID-19 pandemic. This was in line with other studies that suggested motivation-boosting tactics and strategies can promote substance users’ treatment participation and retention, as well as good treatment outcomes such as decreased alcohol and drug use, a higher abstinence rate, and successful referral to treatment [42]. The Transtheoretical Model is the most commonly used model for operationalizing patient motivation to change in drug use therapy. This Stages of Change model depicts categories that patients tend to fall into when making significant changes in their lives, such as entering treatment, and it can be viewed as a continuum along which an individual can move toward long-term or permanent change, such as reducing or even stopping substance use [43].

There are several limitations that should be highlighted in this study. The main limitation of this study was that it used a cross-sectional study design in which the study of causality or causal relationship between the studied factor may not be determined accurately. The other possible limitation includes errors of representation, specifically coverage errors. Coverage errors are usually caused by using a survey sampling frame that does not include all members of the population being researched, or alternatively, by using methods that do not provide all members of the population some chance of being sampled. This risk of falling into a prospective sample frame is linked to drug use behaviors in some situations; substance use research may be particularly sensitive to coverage error [44]. In the case of our study, most of the sample was from the UMMC addiction/substance psychiatry clinic and most were heroin users who came for methadone replacement therapy. Due to this factor, the studied population may not be as heterogenous as it should be, which affects certain aspects of the study results. This is also because the sampling method use was non-probability, convenient sampling. However, many other substance-related studies have used this method as a sampling technique. When research questions focus on certain populations thought to be at higher risk for substance use and misuse, non-probability samples are typically used. There are many well-known non-probability, or convenience sample designs that are frequently employed in practice [44].

## 5. Conclusions

With the ongoing pandemic, many aspects of society are affected, including substance users. Hence, addiction medicine as well as medical practitioners need to adapt to mitigate a possible secondary pandemic of substance use. Although this study aimed to determine the changes in alcohol and illicit substance use during COVID-19 and the possible association with depression and anxiety levels as well as coping mechanisms among PWUD in a cross-sectional way, it gives a rough idea of the significant changes among substance users during the pandemic and of the psychological and other factors that influenced this. Although there was no obvious significant association between depression/anxiety status and coping mechanism with the increase of substance use among PWUD, the prevalence of PWUD exhibiting depressive and anxiety symptoms as well as poor coping mechanisms should be taken seriously, and other factors such as isolation status and intention to seek professional support for drug counseling and treatment appear to be significantly associated with the pattern of substance use among PWUD. Other measures such as proper screening and frequent monitoring of their anxiety and depressive symptoms could be one way to enhance their rehabilitation and possibly lead to abstinence from their substance usage in the future, noted as one of the significant associating factors of reducing substance usage during the pandemic in our study.

By identifying these changes and how they affect PWUD, it is hoped that more actions and preventive measures will be taken to provide treatment and rehabilitation for substance users, especially during future pandemics. Though the COVID-19 pandemic is moving towards endemic status, it is crucial that a more comprehensive and all-around approach be taken in the future.

## Figures and Tables

**Figure 1 healthcare-10-01366-f001:**
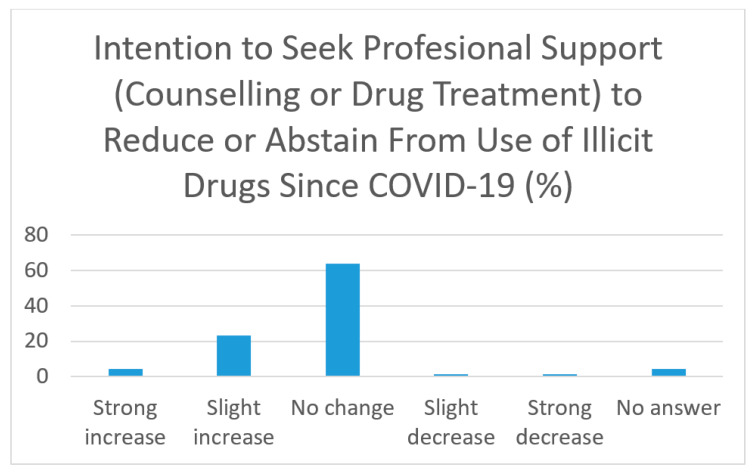
Intention to seek professional support (counselling or drug treatment) to reduce or abstain from use of illicit drugs since COVID-19 restrictions.

**Table 1 healthcare-10-01366-t001:** Sociodemographic data, isolation status, and types of substances used before the COVID-19 pandemic period/restrictions (*n* = 130).

Variables		N	%
**Age (mean, SD)**	Years	45.98	9.786
**Gender**	Male	109	83.9
	Female	21	16.1
**Place of Living**	City	125	96.2
	Town	3	2.3
	Village	2	1.5
**Isolation Status**	Physical and Home Isolation	63	48.5
	Physical Isolation	32	26.6
	Physical, Home Isolation, and Quarantine	8	6.2
**Types of Substances Used Prior to COVID-19 Pandemic Period/Restrictions (Before January 2020)**	Multiple Substances	55	42.3
	Heroin	25	19.2
	Alcohol	13	10
	Amphetamine-Type Stimulant	11	8.5
	Cannabis	6	4.6

**Table 2 healthcare-10-01366-t002:** General use of illicit substances/alcohol pattern during COVID-19 restrictions/pandemic period.

Variables		N	%
**General Use of Illicit Substances/Alcohol During COVID-19 Restrictions**	Have Not Used/Stop Taking The Illicit Substances/Alcohol	47	36.2
	Increased Usage	34	26.2
	Decreased Usage	26	20
	Used The Same Amount of Illicit Substances/Alcohol	19	14.6
	Others	4	3.08

**Table 3 healthcare-10-01366-t003:** Types of substances used during COVID-19 pandemic period compared to prior to COVID-19 pandemic period.

Types of Substances Used Prior to COVID-19 Pandemic Period/Restrictions (before January 2020)	N	%	Types of Substances Used during COVID-19 Pandemic Period (after January 2020)	N	%
Multiple substances	55	42.3	Multiple substances	19	14.6
Heroin	25	19.2	Heroin	16	12.3
Alcohol	13	10	Alcohol	16	12.3
Amphetamine-type stimulant	11	8.5	Amphetamine-type stimulant	16	12.3
Cannabis	6	4.6	Cannabis	9	6.9

**Table 4 healthcare-10-01366-t004:** Pattern of usage for each of the substances.

Types of Substance	Pattern of Usage	N (%)
Heroin	Increased	5 (3.8)
	Decreased	11 (8.5)
	**Stopped**	**27 (20.8)**
	Not changed	26 (20)
Alcohol	Increased	12 (9.2)
	Decreased	4 (3.1)
	Stopped	6 (4.6)
	**Not changed**	**13 (10)**
Amphetamine	Increased	10 (7.7)
	Decreased	6 (3.8)
	Stopped	13 (10)
	**Not changed**	**18 (13.8)**
	Started	2 (1.5)
Cannabis	Increased	6 (4.6)
	Decreased	5 (3.8)
	**Stopped**	**18 (13.8)**
	Not changed	16 (12.3)

**Table 5 healthcare-10-01366-t005:** Changes in purity, price, and amount of illicit substances/alcohol obtained since the outbreak of COVID-19.

Substances on Market	Changes in Substances Obtained	N (%)
Purity	Increased	1 (0.8)
	**Decreased**	**47 (36.2)**
	Same	24 (18.5)
	**Others**	**58 (44.6)**
Price	Increased	13 (10)
	Decreased	7 (5.4)
	**Same**	**52 (40)**
	**Others**	**58 (44.6)**
Amount	Increased	1 (0.8)
	Decreased	14 (10.8)
	**Same**	**58 (44.6)**
	**Others**	**57 (43.8)**

**Table 6 healthcare-10-01366-t006:** Correlation between anxiety and depression scores and different coping mechanisms among people who used drugs/alcohol (PWUD).

Coping Strategy	Anxiety		Depression	
	Correlation, r	*p*-Value	Correlation, r	*p*-Value
Problem focused	0.18	**<0.05**	0.157	0.07
Emotion focused	0.11	0.224	0.096	0.287
Dysfunctional	0.23	0.804	0.035	0.705

**Table 7 healthcare-10-01366-t007:** Simple logistic regression between increased use and variables that are significantly associated with increased use in bivariate analysis.

Variable	Increased Use		No Increased Use		Crude OR (95% CI)	*X2 Stat (df)*	*p*-Value
	N (%)	Mean ± SD	N (%)	Mean ± SD			
Gender					**1.96 (0.73, 5.29)**	1.74	0.18
Male	26 (76.5%)		83 (86.5)				
Female	8 (23.5)		13 (13.5)				
Age		43.68 + 8.25		46.79 + 10.19	0.96 (0.92, 1.00)	2.68	0.11
Location/Place of living						3.10	0.99
Urban	34 (100)		91 (94.8)				
Rural	0 (0)		5 (5.2)				
Isolation status					**2.11 (0.92, 4.90)**	3.23	0.08
Isolated	24 (70.6)		51 (53.1)				
Not isolated	10 (29.4)		45 (46.9)				
Depression					0 (0, 0)	6.35	0.99
Yes	0 (0)		10 (10.4)				
No	34 (100)		86 (89.6)				

**Table 8 healthcare-10-01366-t008:** Multivariable logistic regression between variables in the simple logistic regression and increased use.

Variable	Increased Use	No Increased Use	Adjusted OR (95% CI)	*X2 Stat (df)*	*p*-Value
	N (%)	N (%)			
Isolation status			**2.63 (1.03, 6.66)**	**5.39**	**0.04**
Yes	24 (70.6)	51 (53.1)			
No	10 (29.4)	45 (46.9)			
Depression			0.725 (0.31, 1.72)	0.81	0.47
Yes	0 (0)	10 (10.4)			
No	34 (100)	86 (89.6)			

Model adjusted for age, gender, and place of living.

**Table 9 healthcare-10-01366-t009:** Multivariable logistic regression for intention to seek professional support against reduced use among PWUD.

Variable	Reduced Use		No Reduced Use		Adjusted OR (95% CI)	*X2 Stat (df)*	*p*-Value
	N (%)	Mean ± SD	N (%)	Mean ± SD			
Intention to seek professional support					**4.42 (0.29, 67.50)**	**13.67**	**<0.01**
Increase	28 (41.2)		9 (16.1)				
No increase	40 (58.8)		47 (83.9)				

Model adjusted for age, gender, and depression status.

**Table 10 healthcare-10-01366-t010:** Multivariable logistic regression for dysfunctional coping against no change in pattern of use among PWUD.

Variable	No change in Pattern of Use		Change in Pattern of Use		Adjusted OR (95% CI)	*X2 Stat (df)*	*p*-Value
	N (%)	Mean ± SD	N (%)	Mean ± SD			
Dysfunctional coping					**3.872 (1.18, 12.65)**	**4.686**	**0.025**
Yes	6 (30)		13 (11.8)				
No	14 (70)		14 (88.2)				

Model adjusted for age, gender, and depression status.

## Data Availability

The data presented in this study are available on request from the corresponding author. The data are not publicly available due to privacy and ethical issues.

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
