# Peer review of "The Pattern of Substance Use among People Who Use Drugs (PWUD) Receiving Treatment at University Malaya Medical Centre (UMMC) during the COVID-19 Pandemic and the Associated Factors"

_healthcare, 2022, doi:10.3390/healthcare10081366_

Round 1

Reviewer 1 Report

The paper deals with the issue of substance use trends during the covid epidemic in Malaysia. Some major flaws can de detected and spotted out as follows:

  1. Background and rationale: it is not clear why authors decided to focus on depression and anxiety as variables related to substance use during the covid epidemic, instead of extending the evaluation to all mental health dimensions, using possibly a general instrument for the assessment of psychopathology.
  2. Inclusion and exclusion criteria: authors specify that all patients with substance use disorder are included, but then also add substance abuse (which should be included already) and eventually add recreational use. Exclusion criteria feature people with mental instability, which is in contradiction to the inclusion of substance use disorders themselves, but also with the intent to evaluate levels of anxiety and depression. It may be that authors aim at evaluating substance use in people without dual diagnosis, but this is not specified, and exclusion of psychiatric conditions is suggested very vaguely (mental instability). Since mental instability may be a correlate of recreational use too, and no specific exclusion of psychiatric diagnoses is defined, it is not clear where do people with depressive and anxiety symptoms fall (should they be excluded when assessed as depressed anxious ? should they be included since their diagnoses are induced by environmental factors ?). On one hand, it is not clear how distinctions ca be made about primary and secondary syndromes, but on the other hand contradictions intermingle.
  3. The self report for substance use is not an ideal tool to assess substance use, it may have different implication for licit or illicit drugs, and underrating is possible, without the certainty that it is systematic and covers up all diagnostic groups to the same extent.
  4. Diagnoses are not defined, and Substance use disorders are assessed on an unclear basis (DSM 5 ?)
  5. One variable called “intention to seek help” remains unclear: it is a thought, a behavior, a perception of danger which induces self-criticism about being in need of help ? In fact, it seems that people who “intend to seek help” tend to reduce their substance use, which would mean they tend to go for a solution by themselves (or is it that because they have actually applied for treatment ?)

On the whole, these flaws affect the quality of findings. The study design itself is partial, contradictory and arbitrary with respect to the inclusion and or exclusion (overlapping) of some psychiatric aspects. The findings about the trends during the covid epidemic may be themselves acceptable, but we are afraid they are of little significance.

The paper cannot be accepted in the present form.

Instruments cannot be changed, but the study design may be revised, and the data analysis re-proposed with different aims and conclusions.

Author Response

Hi, thank you for the review.

  1. The purpose of this study is to focus on depression and anxiety, as these are some of the most common comorbidities linked with substance use, however, there was no clear causal relationship between them. I'll add some other literature review in the intro/background
  2. I agreed with this point. I should put the inclusion criteria as anyone who uses alcohol/illicit substances. Regarding the exclusion criteria of mentally unstable, it means whether the respondent in severe/acute episode of psychiatric illnesses
  3. The tool that was used for this study (mini-EWSD COVID 19) is just a survey questionnaire which not assess any severity or any scale about the substance use 
  4. This would be amended in the inclusion criteria 
  5. Intention to seek treatment is one of the survey questions in the mini-EWSD COVID-19 questionnaire. It's just the respondent's feelings/thoughts to seek help for their substance use which may or may not be followed by treatment-seeking behavior after that. 

Reviewer 2 Report

The manuscript shows very important problem, but it has to be corrected and changed.

The procedure selected of participants group is questionable and should be explained by the authors.

Introduction

Accepted

Material and Methods

It is nesesery describe that this study was analyzed declarative and retrospective opinion of participants about use drug and alcohol.

The manuscript should be check and change in edition of text - necessary to remove unnecessary spaces and dots, there is editorial chaos.

Lines 110-114 describe is not understandable: how were recruted patients? In hospital, by physical examination, mental state examination (by used criteria of ICD-10, and DSM-5)? Do they recruted by web questionnaire, only? If recrutation was online, would be difficulty receive inclusion and  exclusion. It should be carefully describe and precise. I have doubts who gave responses online.    

Line 117 – in this places is repeated describe of questionnaires, scales should be precise described i.e authors, validation authors in Malasia

Line 124 – authors wrote: „Subjects who uses substances including substance use disorder, substance abuse or recreational use.” (in Inclusion criteria) or „Subjects who are physical and mentally unstable” (in exlusion criteria) – should be describe, what did authors do, if the study was realized online?

Line 145 – authors showed „The questionnaire is also anonymous…...” and in the criteria inclusion or exclusion authors wrote, that was needs consent of participants – It should be explain, How did authors obtain consent?  

Line – 144 this sentense „All the information given is entirely confidential and will be kept safely.” – should be relocation to the part of Materials and Methods.

Line 157 – please added questionnaire’s authors

Line 197 – authors wrote „Physical isolation, Home Isolation and Home Quarantine…” – should be definated in the part of Materials and Methods

Line 201 –  authors wrote „…prior to COVID-19 pandemic period” – it should be clarify and described  in the part of Materials and Methods. What range of period was used?  

Results

In order to better describe the results, data (for prior to and restriction) should be included in the same table.

Line 202-205 and 208-211 – repeating elements of concepts, unnecessarily

Line 234 – table 4: I do not understand number of increased, decreased, stopped and not changed – it is more than it was showed the drug users in table 3; e.g for the heroin - 66 (table 4) vs 16 (table 3).

Line 250 – Is the table 5 necessary in this a manuscript? This data are not directly connected with the aim of study.

Line 307 – data in the table 7 are disordered e.g. comma, inappropriate places after comma in these variables.

Line 339 – this sentence „*Model adjusted for age, gender and depression status” should be below table 7?  

Discussion

Line 475 and 477 – the pejorative words alcoholism, alcoholics have to corrected

Line 477 – „alcohol addicted” should be change to „alcohol dependence”

Author Response

Materials and method

line 110-114 : patients were recruited in the hospital, mostly in the psychiatry/addiction clinic setting and screened by the authors or other trained doctors in the clinic for inclusion and exclusion criteria before including the patient in this study.

line 117 : noted and will do the needful.

line 124 : what the author mean was all the patients/respondent who use substances/alcohol, can be included in this study ( by giving example of substance use disorder, substance abuse or recreational use). For the exclusion criteria of physically and mentally unstable, although this is an study using online platform, the patients/respondent were recruited via face to face approach by the author (using his own computer for the patient to use to answer the questionnaire) or by the trained doctors in the clinic to carefully selected the interested patient using the inclusion and exclusion criteria provided. 

line 145 : the informed consent was included at the front page of the online platform before the patients starting to answer the questions. There will be a tick box of yes/no there. If the patient answered no, which means disagree to participate in the study, they will not be included in this study. 

line 144 : noted and will do the needful. 

line 157 : noted and will do the needful. 

line 197 : noted and will do the needful. 

line 201 : noted and will do the needful. 

Results

Agreed for the combination of the tables for substance used prior and during COVID-19 

Noted regarding the repetition of elements of concept.

Line 234 (table 4) : the numbers of increase, decrease, stopped and not changed of use were higher in each substances in this table compared to their respective substances in table 3 is because most of the patients uses multiple substances, which causes overlaps of the substance which explained the discrepancies of the numbers. 

line 250 : table 5 was included although it is not related to the aim of this study because it gave general and important idea of possibilities of the market changes causing the change of pattern in substance use during COVID-19

line 307 : noted and will do the needful. 

line 339 : model adjusted for age, gender and depression status were meant for the confounding variables in the multivariable logistic regression.

Discussion

Noted and will do the needful. 

Reviewer 3 Report

Thank you for the opportunity to review this manuscript, which I read with great interest both because of the importance of the topic and the subject matter, which I myself deal with in my scientific research work.

Introduction: good, aim: stated, material and methods: described in detail and sufficiently, as is the results section. The interesting and extensive discussion and the conclusions, in which the researchers mainly point out the limitations, are basic and properly prepared chapters.

In my opinion, the size of the research sample is small, but due to the specificity of the research group and the specific time of the research this is acceptable. However, the results of this study should not be generalised, but the authors have drawn attention to the limitations of the research presented.

Overall, the manuscript is interesting and meets the criteria of a good scientific paper.

However, I have a few comments:

Substantive comments: 

As a sociologist, I must draw attention to the misuse of the term "social distance" (line 41, 112). In the sciences - not only in sociology - the term "social distance" is used to describe differences between social positions (social stratification). Even the World Health Organization points this out, recognising that it is physical distance, not social distance, that prevents the transmission of infections. See Tangermann V., 2020: It is officially time to stop using the phrase 'social distance'. "Science Alert").

I also suggest not to use the stigmatizing terms alcoholism (line 475) alcoholics and drug addicts (line 477). Instead, please use more euphemistic terms, as in the rest of the article.

In my opinion, some values - % and N (especially in 3.2 and 3.5) - should be written without brackets. It is possible to write without brackets the number of respondents, i.e. that it is N, and give the percentage in brackets. This solution is more readable and correct. Alternatively, you can proceed as in section 3.7, where the data are legible and correctly written.

Technical notes: 

Numerous technical errors, especially punctuation, blank spaces in places, missing parts or ends of sentences, improperly started brackets, etc. (chapter titles and many places in the manuscript).

Incorrect numbering: missing section 3.4.

Please also adapt the References section as required by the Journal.

Author Response

Thank you for your review.

Noted regarding the terms of the social distance, will do the needful. 

Noted regarding the use of stigmatizing words, the author will change it. 

Noted regarding the technical errors and reference section as required by the journal. The author will do the needful. 

Round 2

Reviewer 1 Report

ok, amended.

Author Response

Okay, thank you very much for your review.

Reviewer 2 Report

Dear Authors,

The authors took into account the suggestions of reviewer but:

- in the results should be check and change spelling and text editing, especially the tables. The checking of text should include e.g: number places after comma, % (currently is mixed up). The text has to be orderly.

- statistical methods should be refill diffrent used methods e.g Logistic Regression

Best regards

Reviewer

Author Response

Hi, noted and thank you for your review. I already amended and corrected the necessary.